# Comparison of Glucose Metabolizing Properties of Enterobacterial Probiotic Strains In Vitro

**DOI:** 10.3390/nu16162677

**Published:** 2024-08-13

**Authors:** Jules Balanche, Emilie Lahaye, Lisa Bremard, Benjamin Thomas, Sergueï O. Fetissov

**Affiliations:** Regulatory Peptides-Energy Metabolism and Motivated Behavior Team, Neuroendocrine, Endocrine and Germinal Differentiation and Communication Laboratory, Inserm UMR1239, University of Rouen Normandie, 76000 Rouen, France; jules.balanche@univ-rouen.fr (J.B.); emilie.lahaye1@univ-rouen.fr (E.L.); lisa.bremard@etu.univ-rouen.fr (L.B.); benjamin.thomas10@univ-rouen.fr (B.T.)

**Keywords:** glucose metabolism, probiotics, type 2 diabetes, bacterial glucose transporters, gut microbiota

## Abstract

Before the absorption in the intestine, glucose encounters gut bacteria, which may serve as a barrier against hyperglycemia by metabolizing glucose. In the present study, we compared the capacity of enterobacterial strains to lower glucose levels in an in vitro model of nutrient-induced bacterial growth. Two probiotic strains, *Hafnia alvei HA4597* (*H. alvei*) and *Escherichia coli* (*E. coli*) *Nissle 1917*, as well as *E. coli K12*, were studied. To mimic bacterial growth in the gut, a planktonic culture was supplemented twice daily by the Luria Bertani milieu with or without 0.5% glucose. Repeated nutrient provision resulted in the incremental growth of bacteria. However, in the presence of glucose, the maximal growth of both strains of *E. coli* but not of *H. alvei* was inhibited. When glucose was added to the culture medium, a continuous decrease in its concentration was observed during each feeding phase. At its highest density, *H. alvei* displayed more efficient glucose consumption accompanied by a more pronounced downregulation of glucose transporters’ expression than *E. coli K12*. Thus, the study reveals that the probiotic strain *H. alvei HA4597* is more resilient to maintain its growth than *E. coli* in the presence of 0.5% glucose accompanied by more efficient glucose consumption. This experimental approach offers a new strategy for the identification of probiotics with increased glucose metabolizing capacities potentially useful for the prevention and co-treatment of type 2 diabetes.

## 1. Introduction

Glucose is a monosaccharide, playing a key role as a readily available energy source for cell metabolism. The blood level of glucose is tightly regulated by several physiological mechanisms, maintaining it at about 1 g/L in the fasting state. As such, while the supply of new glucose provides its storage in the liver mainly in the form of glycogen, the interplay between the nervous system-regulated hepatic glucose production and the insulin-dependent tissue glucose utilization determines the blood glucose levels [1,2]. Before the arrival of glucose to the liver via the portal vein, it is absorbed from the small intestine following enzymatic hydrolysis of ingested oligo- and polysaccharides by digestive enzymes, such as amylase [3]. The intestinal absorption of glucose is a regulated process involving the SGLT-1 and GLUT-2 molecular transporters, which represent the intestinal barrier against hyperglycemia [4]. In fact, the postprandial concentration of glucose in the gut lumen may exceed more than 10 times its levels in the blood. This is of pathophysiological relevance to the mechanisms of hyperglycemia in type 2 diabetes (T2D), which can be accompanied by increased intestinal glucose absorption [5]. Thus, reducing the glucose charge for the intestinal glucose transporters may represent a useful addition for the prevention and co-treatment of T2D. 

Indeed, the increasing incidence of T2D as a leading cause of several comorbidities requires new approaches for its prevention and therapy [6]. Recently, the gut microbiota was recognized as a new player in many physiological and pathological processes, including the regulation of glucose metabolism and T2D [7,8,9]. The altered composition of the gut microbiome characterizes patients with T2D, suggesting that the microbiome is a valid target for intervention, for example, with probiotics [10,11]. Indeed, the beneficial effects of probiotic supplementation on both glucose and lipid metabolism in T2D were reported [11,12]. Among the molecular mechanisms underlying probiotic bacteria–host beneficial interactions, one can cite anti-inflammatory effects and increased production of short-chain fatty acids (SCFAs) [12,13,14].

Surprisingly, the probiotic bacteria’s capacity to metabolize glucose has not been fully explored in the context of its possible relevance to the host glucose metabolism. Such a capacity should contribute to the glucose metabolizing activity of the indigenous gut microbiota, possibly resulting in a more efficient decrease in glucose in the intestinal lumen. Indeed, bacteria use glucose as an energy substrate for their metabolic needs, such as growth [15]. For this purpose, bacteria express a number of genes encoding nutrient transporters responsible for glucose uptake from the environment [15,16]. A study of glucose utilization by probiotic bacteria is of particular interest during nutrient-induced bacterial growth, i.e., in the conditions imitating the postprandial arrival of nutrients to the gut. Such conditions can be modelized in planktonic bacterial cultures supplemented by nutrients with regular intervals synchronized with day/night cycles, i.e., reflecting daily meals [17]. Our hypothesis was that different probiotic strains may display different glucose-metabolizing activity depending on bacterial density. If confirmed, such data may provide a rationale for the screening of a large number of existing and new probiotic strains for the identification of the most efficient glucose metabolizers. 

Thus, in the present study, we used a “synchronized” model of bacterial growth to evaluate the glucose metabolizing activity of two enterobacterial probiotic strains by continuous monitoring of glucose levels in the culture medium. One tested probiotic was *Hafnia alvei* (*H. alvei*) *HA4597*, a strain that was shown to reduce food intake, body weight and blood glucose levels in obese mice [18,19] and in overweight humans [20]. Another probiotic was *Escherichia coli* (*E. coli*) *Nissle 1917* strain, known for its anti-inflammatory properties [21,22]. As a model of gut commensal enterobacteria, we also studied a laboratory strain of *E. coli K12*. Strains were grown on a standard for enterobacteria Luria Bertani (LB) medium supplemented with 0.5% glucose to model a slight, about five times, increase in the postprandial glucose level in the gut. The choice of enterobacteria instead of Lactobacilli, which are the most common probiotics, was mainly based on the technical feasibility of the experimental approach of continuous glucose monitoring in aerobic conditions. Moreover, as discussed below, both *H. alvei* and *E. coli Nissle* probiotic strains may display anti-diabetic properties.

## 2. Materials and Methods

### 2.1. Synchronized Bacterial Culture

To model the intake of two daily meals in humans, we used an in vitro model called here “synchronized” culture. The first stage consists of preparing a preculture for each of the three strains from lyophilized or frozen stocks. Three bacterial strains, *E. coli K12* (from the Rouen University laboratory collection), *H. alvei HA4597* (Enterosatys^®^, TargEDys SA, Longjumeau, France) and *E. coli Nissle 1917* (Mutaflor^®^, Ardeypharm GmbH, Herdecke, Germany), were grown first at 37 °C in 50 mL of LB medium supplemented or not with 0.5% glucose in an Erlenmeyer flask at 2 g overnight. After one night, bacterial growth was measured at an optical density (OD) at λ = 595 nm by a spectrophotometer. A scale-up step is then carried out by inoculating at 10^6^ CFU/mL in an Erlenmeyer flask containing 200 mL of LB medium supplemented or not with 0.5% glucose. After the preculture, the bacteria were given a new LB medium with 8 h intervals after passages (P) P1, P3 and P5 and 16 h intervals after P2 and P4 during 3 consecutive days. During the cultivation, a bacterial sample (~1 mL) was collected, and the bacterial growth was measured as an OD at λ = 595 nm by a spectrophotometer every hour after the 1st provision of LB medium; then, this sample was stored at −20 °C. At the end of each 8 h or 16 h feeding cycle, bacteria were centrifuged for 5 min at 2800× *g* at room temperature (25 °C) (RT). The supernatants were discarded and replaced by an equivalent volume (~200 mL) of a new LB medium. Three bacterial cultures were grown in Erlenmeyer flasks to monitor glucose consumption by taking samples every hour for 8 h during the day. Finally, the bacterial cultures were grown in parallel in a second support, in a 6-well Costar plate (Merck, Darmstadt, Germany). The cultures were grown in LB medium supplemented or not with 0.5% glucose. The 6-well plate contained 2 mL of bacterial culture per strain. Bacterial growth was measured in real time using a TECAN F200Pro plate reader (Tecan group Ltd., Männedorf, Switzerland), which measures the OD every 5 min, night and day, at λ = 595 nm, while shaking at 2 g and incubating at 37 °C.

### 2.2. Glucose Measurements

Glucose concentration in the culture medium was measured every hour using a colorimetric assay with 3.5 dinitrosalicylic acid (3.5 DNS), a method previously validated [23]. The samples were collected and centrifuged at 8000× *g* for 5 min to pellet the bacteria. An amount of 25 µL of bacterial supernatant was taken, and 25 µL of 3.5 DNS was added. This mixture was then heated at 100 °C for 5 min, and after 5 min, 250 µL of milliQ H_2_O was added. The colorimetric assay was carried out by measuring absorbance at 540 nm, placing 100 µL of solution on a Costar P96 plate, and measuring absorbance using a TECAN F200Pro plate reader.

### 2.3. Expression of Bacterial Glucose Transporters’ mRNA

To measure the expression of the gene of glucose transporters common to the studied bacterial strains, we used reverse transcription–polymerase chain reaction (RT-PCR). From the sequence of these genes retrieved from ‘Uniprot’, primers were designed using ‘Primer BLAST’ (https://www.ncbi.nlm.nih.gov/tools/primer-blast/, accessed on 1 March 2024); their sequences are shown in Appendix A. The primers were then produced by Integrated DNA Technologies (Leuven, Belgium). To validate the PCR primers used, a PCR mix was performed using a dreamTaq mastermix containing dreamtaq polymerase, dreamTaq buffer, dNTPs and 4 mM MgCl_2_; 1.25 µL of H_2_O and 1.25 µL of each of the forward and reverse primers (100 µM); and diluted 10-fold, together with 2.5 µL of bacterial DNA (50 ng/µL). DNA was extracted after bacterial lysis using the QIAamp^®^ Fast DNA kit (Venlo, The Netherlands). The mixes were then amplified using the Thermo Fisher SimpliAmp instrument (Waltham, MA USA). The PCR cycle was performed 35 times and comprised 3 steps: denaturation at 95 °C for 30 s, hybridization at 57 °C for 1 min and amplification at 72 °C for 30 s. For the control of primers, migration of the PCR products was carried out using a 2% agarose E-gel with SYBr gold 2 DNA as the intercalating agent, in parallel with a molecular weight marker. After this step, which confirmed the expected size of the PCR product (Appendix A), we validated the primers by running standard curves with a reaction efficiency of between 90 and 110%. *E. coli K12* and *H. alvei* total RNA was extracted after bacterial lysis using lysis beads and β-mercaptoethanol with the Nucleospin RNA kit (Macherey Nagel, Düren, Germany). After extraction, RNA concentrations were measured using a NanoDrop spectrophotometer. A reverse transcription reaction was performed to generate cDNA with 1 µg of total RNA using the RT Sensifast kit from Meridian Bioscience (Cincinnati, OH, USA). Quantitative PCR (qPCR) was performed on all samples using a BioRad CFX96 Real-Time PCR System and SYBR Green Master Mix (Life Technologies, Carlsbad, CA, USA). The relative levels of gene mRNA expression were estimated by the inverse values of the amplification cycle threshold (CT) for each cDNA sample curve. qPCR was performed to analyze the bacterial gene expression of the different bacterial glucose transporters using a CFX 96 q-PCR instrument (Life Technologies, Carlsbad, CA, USA). The qPCR mix included 6.5 µL of SYBR Green Master Mix (Thermo Fisher Scientific, Waltham, MA USA), 0.39 µL each of the forward and reverse primers (300 nM for each), 3 µL of cDNA from samples (1000 ng/µL), and water to give a total volume of 13 µL. A three-step PCR was performed for 40 cycles. The samples were denatured at 95 °C for 10 min, annealed at 60 °C for 2 min, and extended at 95 °C for 15 s.

### 2.4. Statistical Analysis 

The results were analyzed using GraphPad Prism 9.0.0. For each manipulation, a technical duplicate and biological triplicates were made and represented as mean +/− SEM (standard error of the mean). To demonstrate the normality of the values obtained, the Kolmogorov–Smirnov distribution test was used. Glucose consumption by the three bacterial strains was evaluated using a two-way ANOVA test with a Bonferroni post hoc test to determine the strain effect. RT-qPCR results were analyzed using the Student’s or Mann–Whitney tests, depending on the normality of the values obtained.

## 3. Results

### 3.1. Synchronized Bacterial Cultures

To study the effect of glucose supplementation on bacterial dynamics in synchronized cultures, the growth of each bacterial strain was studied individually by measuring its optical density. Each time the culture medium was renewed, bacterial growth was restarted for all the passages (Figure 1). In a culture medium containing only LB, all three strains displayed similar growth, with *E. coli Nissle* displaying the highest increase (Figure 1a). However, when bacteria were grown in an LB medium supplemented with 0.5% glucose, the bacterial densities of both *E. coli K12* and *E. coli Nissle* strains were lower than those of *H. alvei* (Figure 1b).

### 3.2. Changes in Glucose Concentration in the Culture Medium

The glucose consumption capacity of each bacterial strain was individually assessed in the culture medium during bacterial growth by measuring glucose concentration in the culture medium using the 3.5 DNS method. We found a continuous decrease in glucose concentration during each feeding phase (Figure 2a,b). At relatively low bacterial density, after P1, there was no difference in glucose concentration between the strains (Figure 2a). However, at higher bacterial density after P3 and P5 combined, glucose concentration curves in *H. alvei* and *E. coli Nissle* culture were visually lower than *E. coli K12*, but a significant decrease was present only for *H. alvei* cultures during two last time-points (Figure 2b). As such, after 8 h of growth, the glucose concentration in *H. alvei* culture medium was about 2 times lower than for *E. coli K12*, while *E. coli Nissle* showed intermediate levels. Glucose consumption efficiency was assessed by normalizing the amount of glucose consumed in μg (T0–Tx, for each hour of the passage) and related to the optical density measured by spectrophotometer (c). We found that at the highest bacterial densities during P3 and P5, the amount of glucose consumption was higher for *H. alvei* than for *E. coli K12*.

### 3.3. Dynamics of Bacterial Growth and Glucose Concentration

To compare the dynamics of bacterial growth with glucose consumption, the bacterial density measured for each individual strain was plotted in the same graph overlapping with glucose concentration in the medium. We noticed that a decline in glucose concentration was observed during both the exponential and stationary growth phases of all three bacterial strains at both low and high bacterial density (Figure 3).

### 3.4. Effect of Glucose Supplementation on Glucose Transporters’ Gene Expression 

Based on the significant difference in glucose concentrations between *H. alvei* and *E. coli K12* culture mediums (Figure 2b), as well as superior glucose consumption by *H. alvei* vs. *E. coli K12* (Figure 2c), we analyzed the effects of 0.5% glucose supplementation on glucose transporters’ gene expression in these two strains (Figure 4). The expression levels of the outer membrane porin C (*ompC*), phosphotransferase (*pts*), galactose permease (*galP*) (Figure 4c) and glucokinase (*glk*) (Figure 4d) genes were measured after culturing bacteria with or without 0.5% glucose in LB medium. We found that when *E. coli K12* was grown in the presence of 0.5% glucose, expression levels of all studied genes were decreased as compared with growth conditions without glucose (Figure 4a–d). The same experimental conditions were used for *H. alvei* to study the effect of 0.5% glucose supplementation on maltose transport system permease (*malF*) (Figure 4e), *pts* (Figure 4f), *galP* (Figure 4g) and *glk* (Figure 4h) gene expression. It was observed that when *H. alvei* was grown in the presence of glucose, expression of all the studied genes was decreased as compared with growth conditions without glucose (Figure 4e–h).

### 3.5. Comparison of Gene Expression Levels between Bacterial Strain

To compare the effects of 0.5% glucose supplementation on glucose transporters’ gene expression between *E. coli K12* and *H. alvei* strains, we used the ratios of the cycle threshold gene expression with and without glucose. This analysis revealed a more pronounced glucose-induced decrease in such ratios for *H. alvei* than for *E. coli K12*, including genes encoding for phosphotransferases (*pts*), glucokinase (*glk*), as well as the regulators *sgr* for *H. alvei* and *crr* for *E. coli K12*. However, no significant difference in the expression ratios of the galactose permease (*galP*) gene was observed between the two strains (Figure 5).

## 4. Discussion

The study results have confirmed our working hypothesis that glucose metabolizing activity is both strain-specific and depends on the bacterial density of probiotic bacteria. Since the seminal work of Jacques Monod, a 1965 Nobel prize winner, the bacterial growth dynamic was analyzed to study both the physiology and biochemistry of bacteria. For instance, one of Monod’s postulates stated that the exhaustion of nutrients should be considered as the main limiting factor for bacterial growth [24]. The analysis of bacterial growth in the gut is a challenging task and is currently largely unexplored. A previous study introduced an in vitro model of nutrient-induced bacterial growth in the gut, which was partly validated in vivo in rats [17]. In this model, synchronized with day/night periods, the regular twice daily provision of nutrients to the planktonically grown *E. coli* bacteria induces a relative increase in bacterial density after each ‘feeding’ passage. Thus, the synchronized cultures with different bacterial densities can model different parts of the gastrointestinal tract (GIT) characterized by an increasing gradient of bacterial concentration from the stomach to the large intestine. We should also mention the limitation of this method, which is that it does not reproduce the exact intestinal environment, such as anaerobic conditions, acidity and nutrient composition. In our study, the culture medium was simply selected for the optimal bacterial growth.

In the present study, we used this in vitro model to analyze the effects of nutrient-induced bacterial growth on glucose metabolizing activity of three enterobacterial species, two *E. coli* and one *H. alvei*, which was recently reclassified from Enterobacteriaceae to its own family, Hafniaceae [25]. By using the complex LB nutrient milieu, as expected, we observed the alternation of exponential and stationary phases of growth of all three studied strains of bacteria, which was not dependent on the presence of glucose. Nevertheless, in the presence of glucose, the growth of two *E. coli* species, but not of *H. alvei*, was inhibited starting approximately from the third passage, i.e., at a relatively higher bacterial density than after the first passage. These results suggest that glucose may limit *E. coli* growth in the large intestine, where the highest bacterial density normally occurs.

We found that after the addition of 0.5% glucose to the medium, its concentration gradually decreased during the entire period until the next LB medium and glucose supplementation. Such a decrease was observed during both the exponential and stationary phases of bacterial growth, implying that bacteria continue to actively metabolize glucose even after stopping their division. This finding is of possible relevance to the metabolizing activity of glucose by bacteria in the gut because of the long-lasting, about 6 h duration of the stationary growth phase corresponding to the inter-meal interval [26]. Hence, during the postprandial period, gut bacteria, as exemplified here by some members of the Enterobacterales, appear as robust machinery predictably lowering the glucose levels in their environment. It is noticeable that the concentrations of glucose in the culture medium of high-density *H. alvei* at the end of the ‘feeding’ phases were close to 1 g/L, i.e., to the homeostatic levels of blood glucose in the fasting state. This finding is of possible practical relevance to the present experimental in vitro approach to identify bacterial strains capable of efficiently metabolizing glucose reaching the physiological glucose levels in the blood. While the present study tested some probiotics from only the Enterobacterales order, it would be of practical interest to apply this approach to the Lactobacillales order as the main current source of probiotic bacteria known for their carbohydrate fermentation activities.

A decrease in glucose levels in the culture medium was observed at both low and high densities of all three strains of tested bacteria. Nevertheless, at higher bacterial density, obtained during passages #3 and #5, such a decrease was more pronounced in the *H. alvei* culture. In contrast, similar glucose-lowering dynamics between both studied *E. coli* strains might witness an inherent property of the Escherichia genus. This finding suggests that if present at high density, e.g., in the large intestine, *H. alvei* would have a better glucose metabolizing activity than *E. coli*. This assumption was further supported by calculating the glucose metabolizing efficiency of each of the studied strains in relation to bacterial density. These results also suggest that for the most efficient glucose-metabolizing activity, a higher density of bacteria should be present, which can possibly be achieved only by the long-term regular daily intake of the probiotic strains.

It is interesting that the in vivo putative ability of *E. coli Nissle* to metabolize glucose was tested in an earlier study. The authors found that while oral gavage in mice of *E. coli Nissle* together with glucose improved glycemic response, this effect was independent of the glucose metabolizing activity of bacteria [27]. These in vivo data do not contradict our in vitro results of relatively low metabolizing activity of glucose of this bacterial strain. Nevertheless, a beneficial effect of *E. coli Nissle* on the host metabolism of glucose was observed, suggesting mechanisms independent of the bacterial uptake of glucose. For instance, such an effect can be due to the expression by *E. coli* of the caseinolytic protease B (ClpB) protein, which is a mimetic of α-melanocyte-stimulating hormone (α-MSH), an anorexigenic peptide [28]. *H. alvei* also expresses the ClpB protein, which may underlie its anorexigenic effect in obese mice [18]. In fact, intraperitoneal injection of ClpB or supplementation of ClpB-containing protein extract of *H. alvei* improves glucose tolerance in mice [29]. Furthermore, slightly lower levels of basal glycemia were found in overweight but otherwise healthy, non-diabetic subjects who received daily for 3 months *H. alvei 4597* probiotic strain as compared to the placebo group [20].

A superior glucose metabolizing activity of *H. alvei* as compared to *E. coli* was mainly due to the sustained growth of *H. alvei*, while the growth of both *E. coli* strains in the presence of glucose was inhibited in our model of synchronized bacterial growth. Recent data support the critical role of bacterial density as a key factor in limiting bacterial growth in standard nutrient conditions [30]. To obtain possible insight into the molecular mechanisms of *H. alvei* resilience to elevated levels of glucose, we compared the expression levels of several bacterial genes encoding for glucose transporters between *H. alvei* and *E. coli K12*. As a general phenomenon, we found that both strains significantly reduced the expression of these transporters, most likely as an adaptation to the osmotic stress induced by elevated glucose concentration in the culture medium [31]. Interestingly, a more pronounced downregulation of several glucose transporter gene expressions was found in *H. alvei* than in *E. coli K12*.

In particular, the ompC porin plays an important role in the passage of glucose across the outer membrane of *E. coli*. However, to date, no outer membrane transporters in *H. alvei* have been described as involved in glucose transport from the external environment to the periplasmic space. Galactose permeases are other inner membrane monosaccharide transporters. In the presence of glucose, they are repressed. Other transporters are downregulated in the presence of glucose, such as maltose permeases, which are specific for maltose, a disaccharide composed of two glucose molecules. All these transporters are crucial for optimizing bacterial growth. This is also the case for phosphotranferases, which transport glucose into the bacterial cytoplasm, integrating glucose directly into the glycolysis pathway in the form of glucose-6-phosphate (G6P), and glucokinases, enzymes able to catalyze glucose into G6P. Of interest, it has been noted that the expression of bacterial phosphotransferase systems is altered in patients with T2D [10]. Taken tougher, the expression profile of glucose transporters indicates a more efficient adaptation of *H. alvei* to the osmotic stress, which most likely allows these bacteria to efficiently metabolize glucose and maintain its sustained growth.

Nevertheless, *E. coli* and other members of the Enterobacteriaceae family persist and can even be enriched in the gut microbiota of subjects with TD2, which sometimes contributes to dysbiosis [32,33,34]. The functional significance of such an increase is not yet clear; however, in experimental animals, increasing *E. coli* in gut microbiota was associated with a worse diabetic state [35]. We may speculate that besides *E. coli*-derived endotoxin, inducing chronic inflammation [14], a reduced glucose metabolizing activity of *E. coli* can also contribute to the diabetic state. Nevertheless, *E. coli* is a normal constituent of healthy gut microbiota, suggesting that it may also provide beneficial effects to the host. For instance, increased abundance in the gut microbiota of some enterobacterial members, including *Hafnia*, was associated with lower body mass index [18]. Moreover, an increase in Enterobacteriaceae in gut microbiota was reported after bariatric surgery in patients with obesity and T2D, and it was oppositely associated with meal-induced glycemic response but positively with glucagon-like peptide-1 (GLP-1) secretion [36]. GLP-1 is a key incretin hormone protecting against diabetes [37].

## 5. Conclusions

In conclusion, our results demonstrate different growth and glucose metabolizing activities of three tested strains of enterobacteria in an in vitro model of nutrient provision of gut bacteria supplemented by 0.5% glucose. We show that the probiotic strain *H. alvei 4597* displayed sustained growth and better than *E. coli K12* glucose-metabolizing activity. Thus, our study gives an example of an experimental approach to identify the probiotic strains characterized by intrinsically elevated glucose metabolizing activities. Such strains should be then further validated in animal models of diabetes to determine if they can be useful for the prevention and co-treatment of type 2 diabetes as the first barrier in the gut against hyperglycemia.

## Figures and Tables

**Figure 1 nutrients-16-02677-f001:**
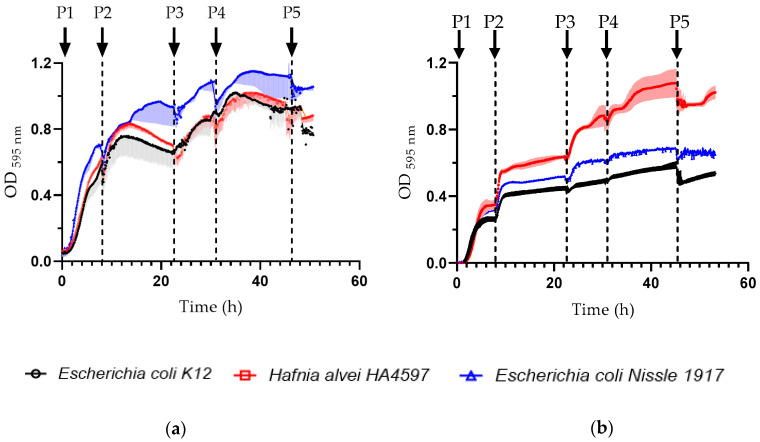
Growth dynamics of three studied enterobacterial strains in synchronized cultures fed by LB medium without glucose (**a**) or supplemented with 0.5% glucose (**b**). Each medium renewal passage (P) is indicated by a black arrow. P1, P3 and P5 are 8 h, and P2 and P4 are 16 h feeding periods. OD_595nm_: optical density at 595 nm.

**Figure 2 nutrients-16-02677-f002:**
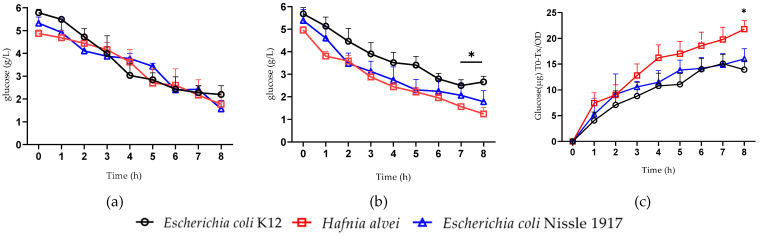
Dynamics of glucose levels in the culture medium of three studied enterobacterial strains at low bacterial density P1 (**a**) and high bacterial density P1 and P3 combined (**b**). Glucose consumption at high bacterial density (**c**). ANOVA, Bonferroni post hoc test * *p* < 0.05. Glucose (µg) T0–Tx/OD: glucose consumption efficiency was assessed by normalizing the amount of glucose consumed (in μg) at each hour (T0–Tx) to bacterial density as measured by optical density (OD).

**Figure 3 nutrients-16-02677-f003:**
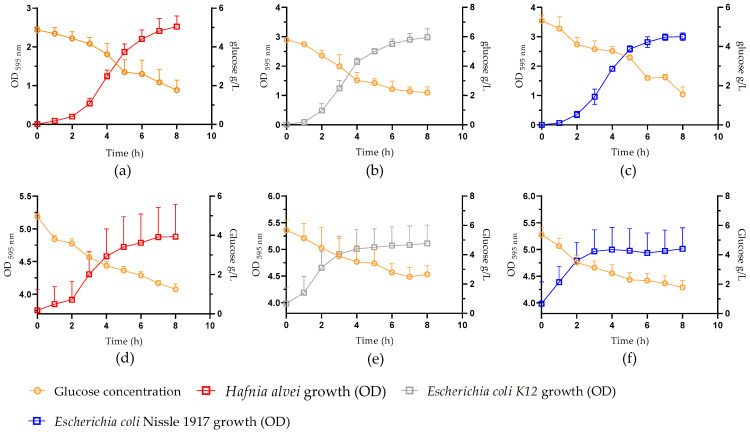
Overlapping dynamics of bacterial growth and glucose concentration in culture medium. Low bacterial density after P1 (**a**–**c**) and high bacterial density after P3 and P5 combined (**d**–**f**). OD_595nm_: optical density at 595 nm.

**Figure 4 nutrients-16-02677-f004:**
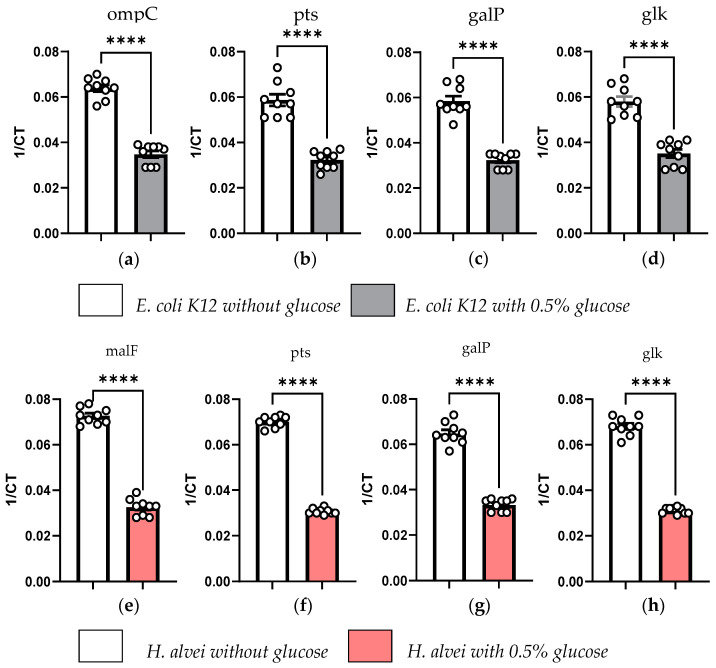
Effect of 0.5% glucose on gene expression in *E. coli K12* and *H. alvei*. As part of a synchronized culture, *E. coli K12* and *H. alvei* were grown in LB medium until passage 3, hour 7. The bacteria were not supplemented with glucose (white bars) or were supplemented with 0.5% glucose (grey and red bars). CT, cycle threshold. Gene expression levels of *E. coli K12* (**a**–**d**) and *H. alvei* (**e**–**h**). **** *p* < 0.0001, Student’s *t*-test (**a**,**d**–**f**) and Mann–Whitney test (**b**,**c**,**g**,**h**). Outer membrane porin C (*ompC*), phosphotransferase (*pts*), galactose permease (*galP*), glucokinase (*glk*) and maltose transport system permease (*malF*).

**Figure 5 nutrients-16-02677-f005:**
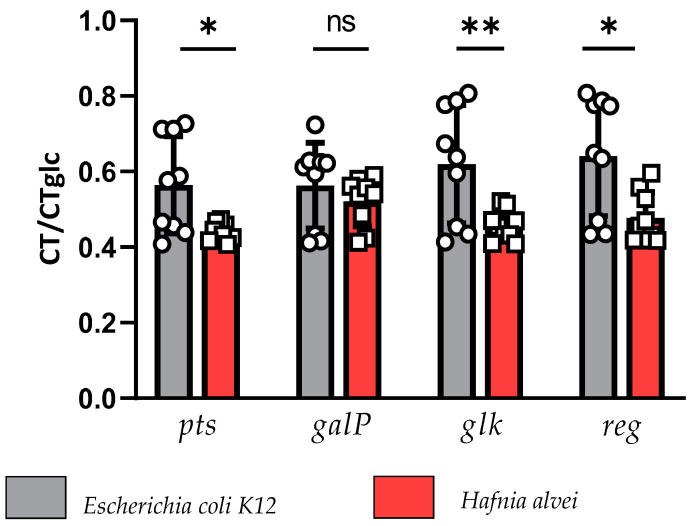
Comparison of the ratios of glucose transporters’ cycle threshold (CT) gene expression without/with glucose (glc) between *E. coli K12* and *H. alvei* strains. ** *p* < 0.01; * *p* < 0.05, Student *t*-test (*pts, glk* and *reg*) or Mann–Whitney test (*galP*). Phosphotransferase (*pts*), galactose permease (*galP*), glucokinase (*glk*), sgr for *H. alvei* and *crr for E. coli* K12 (*Reg*).

## Data Availability

Data are available by request to the corresponding author.

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
