# Peer review of "Comparison of Glucose Metabolizing Properties of Enterobacterial Probiotic Strains In Vitro"

_nutrients, 2024, doi:10.3390/nu16162677_

Round 1

Reviewer 1 Report

Comments and Suggestions for Authors

The data are insufficient to support sustained hypoglycemic properties of the probiotic Hafnia alvei HA4597 in vitro. The paper lacks sufficient data to support the point of view of the paper and proposes to supplement the protein level changes of glucose transporters in different strains in different glucose concentration environments.

1、Line 14 “LB” needs to be full expanded at first mention. Please check the full text.

2、There are some minor mistakes in this manuscript. For example, Line 42, 48 “TD2”; Line 49; Line106 “H2O”; Line 93 “ml”, etc.

3、Section 2.1: Why not choose OD600 to measure bacterial growth? The LB medium used also failed to simulate the real intestinal environment, including acidity and nutrient comparisons.

4、Section 2.2: Please, validate and refine this method.

5、Figure 4 should be given as clear picture.

6、Line 51: Please, change short chain fatty acids (SCFA) to short chain fatty acids (SCFAs); Line 22: Please, change diabetes type 2 to type 2 diabetes; change “p” to “P”. Please check the full text.

7、Please, check the format of tables and images according to journal requirements. The abbreviations used in the tables and figures should be explained in the footnote of tables and/or figure captions to make them standalone.

8、Section 4 - The discussion on results has not been extensively performed, which is crucial to highlight any possible promising applications, especially correlating the experimental data and results obtained by different analytical methods.

9、Revise carefully reference style. For example, volume (number included or not), Journal name (italic or not), etc.

Comments on the Quality of English Language

Moderate editing of English language required

Author Response

The data are insufficient to support sustained hypoglycemic properties of the probiotic Hafnia alvei HA4597 in vitro. The paper lacks sufficient data to support the point of view of the paper and proposes to supplement the protein level changes of glucose transporters in different strains in different glucose concentration environments.

Response. We appreciate the Reviewer’s comments and corrections. We completely agree with the comment, that there are no sufficient data to claim the “hypoglycemic” properties of H.alvei. In fact, to prove this point, we would need an animal or ex vivo model analyzing the blood glucose levels. As stated in the title, we mention only glucose lowering properties of H.alvei in vitro, a conclusion justified by the results.

We also certainly agree that a more deep molecular analyzis of the bacterial strains, including the protein levels of glucose transporters, as suggested by the Reviewer, would be necessary to identify key elements underling the individual strain differences of the glucose metabolism. Our approach, however, by analyzing the glucose transporters’ mRNA, was to provide an example of a molecular process which may overall explain the observed phenomenon of a more sustained in vitro growth of H.alvei in the presence of glucose. These molecular data are somewhat secondary to the in vitro monitoring of glucose levels during bacterial growth which was the primary objective of the study. Adding the protein analysis at this stage will not be feasible due to technical limitations for the specific bacterial proteins antibody generation and validation.

Most importantly, with this study, as we explained in the discussion section, we wanted to provide an experimental approach potentially useful for the identification of new probiotics with increased glucose metabolizing properties. H.alvei is a mere example and, most likely better glucose metabolizing probiotics should exist. To emphasize this point we modified the last sentence in the abstract. Moreover, to take away the focus from H.alvei, but to highlight the study objective, we changed the title of the manuscript to “Comparison of glucose metabolizing properties of enterobacterial probiotic strains in vitro”.

1、Line 14 “LB” needs to be full expanded at first mention. Please check the full text.

Response: done

2、There are some minor mistakes in this manuscript. For example, Line 42, 48 “TD2”; Line 49; Line106 “H2O”; Line 93 “ml”, etc.

Response: done

3、Section 2.1: Why not choose OD600 to measure bacterial growth? The LB medium used also failed to simulate the real intestinal environment, including acidity and nutrient comparisons.

Response: The use of 595 nm OD was due to the specifications of the filter in our TECAN instrument. It is common to use close to 600nm values for measuring bacterial densities. We agree with the limitation of the LB medium, it was used as a glucose free medium optimal for enterobacteria. This limitation was included in the discussion.

4、Section 2.2: Please, validate and refine this method.

Response:  The reference to this method was provided and will be incorporated in the ref list at the final stage of formatting by the reference manager, currently not available because of revising outside of the office.

5、Figure 4 should be given as clear picture.

Response: The figure 4 was replaced with one having better resolution.

6、Line 51: Please, change short chain fatty acids (SCFA) to “short chain fatty acids (SCFAs)”; Line 22: Please, change “diabetes type 2” to “type 2 diabetes”; change “p” to “P”. Please check the full text.

Response: done.

7、Please, check the format of tables and images according to journal requirements. The abbreviations used in the tables and figures should be explained in the footnote of tables and/or figure captions to make them standalone.

Response: It has been checked

8、Section 4 - The discussion on results has not been extensively performed, which is crucial to highlight any possible promising applications, especially correlating the experimental data and results obtained by different analytical methods.

Response: The discussion has been extended as indicated by the highlighted parts.

9、Revise carefully reference style. For example, volume (number included or not), Journal name (italic or not), etc.

Response: we have used a reference manager with the mdpi style, we will additionally check the formatting at the final stage when the ref list will be updated.

Reviewer 2 Report

Comments and Suggestions for Authors

This study provides interesting information. However, the following points should be addressed before publication.

1. The difference in glucose responses appear to be slight between control (E. coli K12) and the probiotic (E. coli Nissele 1917) (Fig 2 and 3). Can this difference be sufficiently explained by the presence and absence of probiotic property? How do authors respond to this criticism.

2. The environment of gut lumen is generally anaerobic conditions. Therefore, further study is necessary to investigate the effects of probiotics under anaerobic conditions. 

3. The rationale for 0.5% glucose should be explained.

Author Response

This study provides interesting information. However, the following points should be addressed before publication.

  1. The difference in glucose responses appear to be slight between control (E. coli K12) and the probiotic (E. coli Nissle 1917) (Fig 2 and 3). Can this difference be sufficiently explained by the presence and absence of probiotic property? How do authors respond to this criticism.

Response: We thank the Reviewer for the comments. Similar glucose lowering dynamics between both studied E.coli strains might witness an inherent property of the Escherichia genus. The growth of both strains was also similarly affected by glucose. We added this clarification to the discussion.

  1. The environment of gut lumen is generally anaerobic conditions. Therefore, further study is necessary to investigate the effects of probiotics under anaerobic conditions. 

Response: This is a very good point, it was now included as a limitation of our in vitro model. In fact, for the technical purposes it was easier to monitor glucose levels using aerobic conditions in facultative anaerobe species.

  1. The rationale for 0.5% glucose should be explained.

Response: supplementation with 0.5% glucose was used to model a slight, about 5 times, increase of the postprandial glucose level in the gut. This explanation was included in the introduction.

Reviewer 3 Report

Comments and Suggestions for Authors

nutrients-3129878-peer-review-v1

The current paper provides an interesting research proposal, where probiotics can be applied as consumer for the glucose in the GIT and by this approach to help in the reduction of the glucose and help in the control of diabetes. The research proposal is interesting and original, however, maybe authors can add a few lines for the choice of that specific probiotics. Why Hafnia alvei and E. coli will be suitable for such a job? Why not some other species? A few lines will need to be provided in this this direction.

Even constructed as a simple instrumental project, the work giving interesting results that showing that some bacterial species applied as probiotics may have effect in the reduction of glucose levels in the GIT and by this way, help in the reduction of blood glucose levels. Maybe in addition (as next step of the research), experiments with animal model can confirm the authors hypostasis. Will be interesting to see if the mentioned strains will have a similar behavior when exposed to the real GIT conditions.

Ln91: In this and similar occasion, centrifugation conditions need to be presented as g force, not as rpm. Please, correct. Please, when you mention room temperature, please, specify it as oC.

For the PCR reactions, please, provide appropriate molarity of applied solutions and concentration of applied DNA/RNA. Please, be more specific.  DNA/RNA isolation was done at what stages?

Maybe authors can consider organizing their ideas about using figures 1, 2 and 3, since in fact figure 3 can be regarded as combination of previous 2. Maybe Fig 3 can be removed to the supplementary material.

Better quality of figure 4 need to be provided.

Author Response

The current paper provides an interesting research proposal, where probiotics can be applied as consumer for the glucose in the GIT and by this approach to help in the reduction of the glucose and help in the control of diabetes. The research proposal is interesting and original, however, maybe authors can add a few lines for the choice of that specific probiotics. Why Hafnia alvei and E. coli will be suitable for such a job? Why not some other species? A few lines will need to be provided in this this direction.

Response: We thank the reviewer for the appreciation of our study and the comments. To explain the choice of tested probiotic strains, the following sentence was included in the introduction.  “The choice of the Enterobacteria instead of ex Lactobacilli, which are most common probiotics, was mainly based on the technical feasibility of the experimental approach of continuous glucose monitoring in aerobic conditions. Moreover, as discussed below, both H.alvei and E.coli Nissle probiotic strains may display anti-diabetic properties.”

Even constructed as a simple instrumental project, the work giving interesting results that showing that some bacterial species applied as probiotics may have effect in the reduction of glucose levels in the GIT and by this way, help in the reduction of blood glucose levels. Maybe in addition (as next step of the research), experiments with animal model can confirm the authors hypostasis. Will be interesting to see if the mentioned strains will have a similar behavior when exposed to the real GIT conditions.

Response: Thank you for this comment, in fact, we see our study as a new in vitro approach to identify the best glucose metabolizing probiotic strains, which activity then should be validated in vivo. Now, we clarified this part in the conclusion. With regard to H.alvei and E.coli Nissle strains they both were used in humans and/or animals showing modest anti-diabetic activity, these data were mentioned in the discussion.

Ln91: In this and similar occasion, centrifugation conditions need to be presented as g force, not as rpm. Please, correct. Please, when you mention room temperature, please, specify it as oC.

Response: Done

For the PCR reactions, please, provide appropriate molarity of applied solutions and concentration of applied DNA/RNA. Please, be more specific.  DNA/RNA isolation was done at what stages?

Response: DNA was extracted after bacterial lysis using LyseBlue reagent. The RNA was extracted after bacterial lysis using lysis beads and β-mercaptoethanol. PCR was performed using a dreamTaq mastermix containing (dreamtaq polymerase, dreamTaq buffer, dNTPs and 4 mM MgCl2 --> thermofisher (no further information available). The PCR primers were at a concentration of 100µM, the quantity of DNA for the PCR was 50 ng/µL and the extracted arn was transformed into cDNA at a concentration of 1000 ng.

Maybe authors can consider organizing their ideas about using figures 1, 2 and 3, since in fact figure 3 can be regarded as combination of previous 2. Maybe Fig 3 can be removed to the supplementary material.

Response: In our view, Figure 3 although derived from the data shown in Figs 1 and 2 is necessary to be shown to clearly demonstrate the temporal overlapping dynamics of bacterial growth and glucose concentration. We discussed this point and would like to keep it as a main figure.

Better quality of figure 4 need to be provided.

Response: done

Round 2

Reviewer 1 Report

Comments and Suggestions for Authors

Accept

Reviewer 2 Report

Comments and Suggestions for Authors

This manuscript was well revised, and appears to be appropriate for publication.